# Effect of a Preoperative Proton Pump Inhibitor and Gastroesophageal Reflux Disease on Postoperative Nausea and Vomiting

**DOI:** 10.3390/jcm9030825

**Published:** 2020-03-18

**Authors:** Young Suk Kwon, Jun Woo Choi, Ho Seok Lee, Jong Ho Kim, Youngmi Kim, Jae Jun Lee

**Affiliations:** 1Department of Anesthesiology and Pain Medicine, Chuncheon Sacred Heart Hospital, College of Medicine, Hallym University, Chuncheon 24253, Korea; gettys@hallym.or.kr (Y.S.K.); guacamole@hallym.or.kr (H.S.L.); poik99@hallym.or.kr (J.H.K.); 2Institute of New Frontier Research Team, Hallym University, Chuncheon 24253, Korea; kym8389@hallym.ac.kr

**Keywords:** proton pump inhibitor, gastroesophageal reflux disease, postoperative nausea and vomiting

## Abstract

Postoperative nausea and vomiting (PONV) are common complications after anesthesia, but no study has considered the effects of a proton pump inhibitor (PPI) and gastroesophageal reflux disease (GERD) on PONV at the same time. Thus, we investigated the effects of a PPI and GERD on PONV. Patients aged ≥18 years who underwent general anesthesia between 2010 and 2019 were enrolled. In total, 202,439 patients were included and 21,361 In a multivariate analysis, the OR for PONV was higher in subjects with GERD (OR, 1.157; 95% CI, 1.032–1.298; *p* = 0.012). The OR was lower for subjects with taking a PPI (OR, 0.890; 95% CI, 0.832–0.953; *p* < 0.0001). In patients without GERD, the incidence of PONV was lower when lansoprazole (OR, 0.801; 95% CI, 0.718–0.894; *p* < 0.0001), pantoprazole (OR, 0.856; 95% CI, 0.748–0.980; *p* = 0.025) and ilaprazole (OR, 0.391; 95% CI, 0.158–0.966; *p* = 0.042) were taken. However, in GERD patients, all PPIs did not show reducing the incidence of PONV. Taken together, the results show that a lansoprazole, pantoprazole, and ilaprazole reduced PONV in patients without GERD, and PPI could not reduce PONV in patients with GERD.

## 1. Introduction

Postoperative nausea and vomiting (PONV) is defined as nausea or vomiting that occurs within the first 24–48 h after surgery [1]. PONV is a common complication after anesthesia and surgery, and affected patients may be more troubled by PONV than postoperative pain [2]. In rare cases, PONV may be associated with suture opening, aspiration of gastric contents, and esophageal rupture [1]. Many reports have cited factors that alleviate or exacerbate PONV [1,3,4,5]. Proton pump inhibitors (PPIs) are occasionally used to reduce pH and the amount of gastric acid before general anesthesia, but they may also have an anti-nausea effect [6]. Gastroesophageal reflux disease (GERD), also known as acid reflux, is a disease in which the stomach contents rise into the esophagus and cause symptoms or complications. GERD can lead to abnormal reflux of gastric contents into the esophagus due to failure of the anti-backflow function of the lower esophageal sphincter (LES) [7]. The gastrointestinal tract can be distended after anesthesia and surgery [8,9], and vomiting is likely to occur in situations where there is dysfunction of the LES, such as GERD. GERD itself can also cause nausea [7]. However, few studies have investigated the effects of a PPI and GERD on PONV, and none has considered their effects on PONV at the same time. Thus, we investigated the effects of a PPI and GERD on PONV in a large retrospective study.

## 2. Materials and Methods

### 2.1. Patients

This study was approved by the Clinical Research Ethics Committee of Chuncheon Sacred Heart Hospital, Hallym University (IRB No. 2020-01-002). All data were obtained from the Clinical Data Warehouse, which includes data from five hospitals at Hallym University Medical Center.

The subjects were ≥18 years of age and had undergone surgery under general anesthesia at one of the five hospitals of Hallym University Medical Center between January 2010 and December 2019. Exclusion criteria included:Reoperation within 24 hunconscious patient;patients treated with a ventilator after surgery;patients with nausea or vomiting before surgery;patients with missing data in their medical records; andpatients with preoperative PPI type changes.

### 2.2. PONV, PPI, GERD, and Other Variables

PONV was defined as nausea or vomiting within 24 h after surgery. In healthy people, the irreversible binding of a PPI to H^+^, K^+^-ATPase results in a high median intragastric pH for 24 h [10]. Therefore, in this study we determined whether subjects were taking a PPI based on the 24-h period before surgery. Indications for PPI use included treatment of GERD, gastric and duodenal ulcer, Helicobacter pylori, and premedication to reduce the risk of aspiration pneumonia before anesthesia in this study. Also, because PPI treatment of GERD for at least 4 weeks is recommended [11], patients diagnosed with GERD within 4 weeks prior to surgery were enrolled as GERD patients. Patients were classified using the 10th International Statistical Classification of Diseases and Related Health Problems. GERD included not only cases of abnormal findings by endoscopy but also those diagnosed with reflux symptoms.

First, we analyzed the effect of GERD and PPI on PONV. We also analyzed known and expected risk factors for PONV, including age, sex, body mass index, anesthesia time, American Society of Anesthesiologists physical status classification, use of N_2_O, anesthetic maintenance drugs, patient-controlled analgesia (PCA), hypertension, diabetes mellitus, heart disease, cerebral stroke, smoking, opioid use during surgery, steroids, antiemetics, neostigmine, antibiotics, laparoscopic surgery, Levin tubes and transfusions [3,4]. Second, the effect of proton pump inhibitors on PONV was analyzed according to presence or absence of GERD. PPIs included were omeprazole, pantoprazole, lansoprazole, rabeprazole, esomeprazole, dexlansoprazole, and ilaprazole, and the effect of each PPI type on PONV was analyzed.

### 2.3. Statistics

Continuous data are presented as mean and standard deviation, and categorical data as frequencies and percentages. For continuous data, *t*-tests were used to compare patients with and without PONV. Categorical data were analyzed using the chi-square test or Fisher’s exact test. The odds ratio (OR) for occurrence of PONV within 24 h after surgery was determined for each variable by logistic regression. Variables selected by forward selection and backward elimination were included in the regression analyses. In the analysis based on presence and absence of GERD, the unadjusted odds ratio and fully adjusted odds ratio for occurrence of PONV according to PPI types were determined. All reported P-values were two-sided, and a *p*-value < 0.05 was considered significant. SPSS software (version 24.0; IBM Corp., Armonk, NY, USA) was used for all statistical analyses. The data set used in our study is provided in Appendix A.

## 3. Results

### 3.1. Study Population

In total, 225,110 patients underwent general anesthesia during surgery between January 2010 and January 2019; 22,671 patients were excluded, and 202,439 patients were included in the analysis. Among the included patients, 21,361 developed PONV within 24 h after surgery; 3537 patients had GERD; 11,111 patients took a PPI before general anesthesia. Table 1 summarizes the demographic characteristics and clinical data of the patients who received general anesthesia during surgery. Significant differences were observed in all variables between the PONV and no-PONV groups (*p* < 0.05), except age (*p* = 0.172), GERD (*p* = 0.264), and Levin tubes (*p* = 0.574).

### 3.2. Odds Ratio for PONV

Figure 1 shows the ORs for the occurrence of PONV of each variable, obtained through simple logistic regression. The unadjusted OR for the occurrence of PONV in subjects with taking PPI was 0.899 (95% confidence interval [CI], 0.843–0.959; *p* = 0.0001) relative to those without taking PPI. The unadjusted ORs for the occurrence of PONV were not significantly different between presence and absence of GERD.

The fully adjusted ORs for the occurrence of PONV of each variable, obtained through a multivariate logistic analysis including all variables, are summarized in Figure 2. The fully adjusted OR for the occurrence of PONV was 1.157 (95% CI, 1.032–1.298; *p* =0.012) when the subject have GERD. The fully adjusted OR for the occurrence of PONV was 0.890 (95% CI, 0.832–0.953; *p* = 0.001) when the subject was taking a PPI. The adjusted ORs for GERD and PPI (i.e., adjusted through backward elimination or forward selection of variables) in the logistic regression are summarized in Table 2.

Table 3 shows types of PPI according to presence and absence of GERD. When patients had GERD, the incidence of PONV was lower when lansoprazole (OR, 0.884; 95% CI, 0.795–0.984; *p* < 0.023), pantoprazole (OR, 0.792; 95% CI, 0.694–0.904; *p* < 0.0001) and ilaprazole (OR, 0.390; 95% CI, 0.159–0.955; *p* =0.039) were taken in univariate logistic analysis. The incidence of PONV was lower when lansoprazole (OR, 0.801; 95% CI, 0.718–0.894; *p* < 0.0001), pantoprazole (OR, 0.856; 95% CI, 0.748–0.980; *p* = 0.025) and ilaprazole (OR, 0.391; 95% CI, 0.158–0.966; *p* =0.042) were taken in multivariate logistic analysis. When patients had GERD, there was no significant difference in the incidence of PONV between doses and non-dose at all types of PPI (Table 4).

## 4. Discussion

Using large data included 202,439 patients, we estimated the effect of GERD and preoperative PPI on occurrence of PONV. When we assessed the odds ratios of occurrence of PONV by GERD and PPI using univariate logistic regression, GERD was not associated with PONV incidence in patients under general anesthesia. In multivariate analysis, however, GERD and PPI were associated with an increase and decrease in occurrence of PONV. Subsequently, we analyzed the effect of each PPI type on occurrence of PONV, and the effect of PPI on the occurrence of PONV differed depending on the presence or absence of GERD.

Many studies have documented the risk factors for PONV among patients who have undergone anesthesia and surgery. Some studies have reported that PONV is associated with both taking a PPI and GERD. Evidence from randomized trials demonstrated that while taking a PPI preoperatively is not associated with the occurrence of PONV [12,13], GERD is a predictor of PONV [14]. Raeder and colleagues reported that esomeprazole was not effective for PONV but could reduce the amount of vomiting [12]. Although their study had included relatively large covariates, it may be difficult to generalize the results because of the high proportion (97%) of women. Female gender is a well-known risk factor for PONV, and this is confirmed in our study. Weilbach and colleagues also reported that prophylactic esomeprazole did not reduce PONV [13]. Their study did not include enough covariates, an extremely large number of women (98.9%), and 43% of all surgeries were gynecological interventions. The results of two studies were in agreement with our findings that esomeprazole did not affect the occurrence of PONV. However, it is difficult to conclude that all PPI does not reduce PONV in patients who have undergone general anesthesia.

PPIs inhibit gastric H^+^, K^+^-ATPase through covalent bonding to cysteine residues in the proton pump [15]. H^+^, K^+^-ATPase is involved in the last stage of acid secretion, and inhibitors of this enzyme are more effective than receptor antagonists for suppressing gastric acid secretion [16]. PPIs may have an anti-nausea effect because receptors in the gastric wall play a role in the feeling of nausea, by mediating input to the nausea-vomiting center in the midbrain received through the vagus nerve [6]. However, GERD patients receiving preoperative PPIs did not reduce the occurrence of PONV in this study. Lansoprazole, pantoprazole and ilaprazole have been shown to reduce the incidence of PONV in patients without GERD in this study. PPIs such as esomeprazole are more effective at controlling gastric acid than first-generation PPIs such as lansoprazole and pantoprazole [17,18]. Other mechanisms besides PPI’s gastric acid regulation may be associated with PONV. More research is needed on these results.

Ilaprazole was associated with the lowest probability of PONV. Most PPIs can result in clinically significant drug interactions that result from the suppression or induction of CYP2C19 as well as CYP3A4 due to other concomitant medications [19]. Midazolam and diazepam, which are metabolized by CYP enzymes, have many clinically significant interactions with inhibitors and inducers of CYP3A4 and 2C19. In addition to pharmacokinetic interactions, benzodiazepines have synergistic interactions with other hypnotics and opioids [20]. Because the metabolic process through CYP3A4/5 [21], ilaprazole may have little effect on the metabolism of benzodiazepine.

GERD is a long-term condition in which the stomach contents rise in the esophagus, causing symptoms or complications [22]. GERD is characterized by symptoms of heart burn, chest pain, regurgitation, and dysphagia, and can also cause nausea or vomiting on its own [23,24]. Anesthesia causes gastrointestinal distension [9], which can increase abdominal pressure and reflux. In turn, reflux is associated with nausea and vomiting [24,25]. An adaptive change in the tone of low esophageal sphincter occurs with a general increase in abdominal pressure [26]. However, the adaptive response to increased intra-abdominal pressure is abnormally low in patients with symptoms of reflux, including GERD patients [27].

We have observed a decrease in PONV in many patient populations during our clinical practice, in contrast to clinical trials. It can be argued that patients enrolled in clinical trials may not represent patients who actually present to the clinic [28]. Our study enrolled more than 200,000 patients; our population may be more representative sample of patients undergoing general anesthesia, referred for the assessment of PONV, than those of a number of randomized controlled trials that previously evaluated the associations of PONV with PPI use or GERD.

The current study reinforces previous findings in several important respects. First, PPI use and GERD are closely associated with PONV. GERD increases the incidence of PONV. While some PPIs reduced the incidence of PONV in patients without GERD, PPI does not reduce the incidence of PONV in patients with GERD. Second, we focused on patients who had undergone general anesthesia, and so reported the risk for PONV according to general anesthetic agents. In addition, covariates included diseases such as hypertension, heart disease, and stroke, which have not previously been associated with PONV.

The major limitation of this study was that the group assignment was not randomized according to PPI use and GERD. The utility of observational studies for evaluating the effect of taking a PPI and GERD on PONV is controversial [29], although several studies have suggested that observational designs are not likely to produce misleading or biased results [30,31,32]. Nevertheless, it should be acknowledged that observational studies can only partially control for confounders, and should only use instruments that can be adjusted accordingly. Other limitations of our study also included a lack of on indications for PPI therapy, the types and amounts of opioids used, and on adjustments to the PCA rate, which are important risk factors for PONV. Lack of information can lead to potential heterogeneity of patient characteristics.

## 5. Conclusions

This study included a large number of patients and many risk factors for PONV, and also controlled for the effects of confounding variables through multivariate analyses Preoperative administration of a lansoprazole, pantoprazole and ilaprazole reduced PONV in patients without GERD. GERD may increase PONV, GERD patients did not reduce the incidence of PONV even when taking PPI. Therefore, our findings provide additional support that GERD is associated with increasing PONV and preoperative esomeprazole is not effective at reducing PONV. However, future prospective, population-based studies are needed to investigate the association between preoperative use of lansoprazole, pantoprazole and ilaprazole, and its impact on reducing PONV.

## Figures and Tables

**Figure 1 jcm-09-00825-f001:**
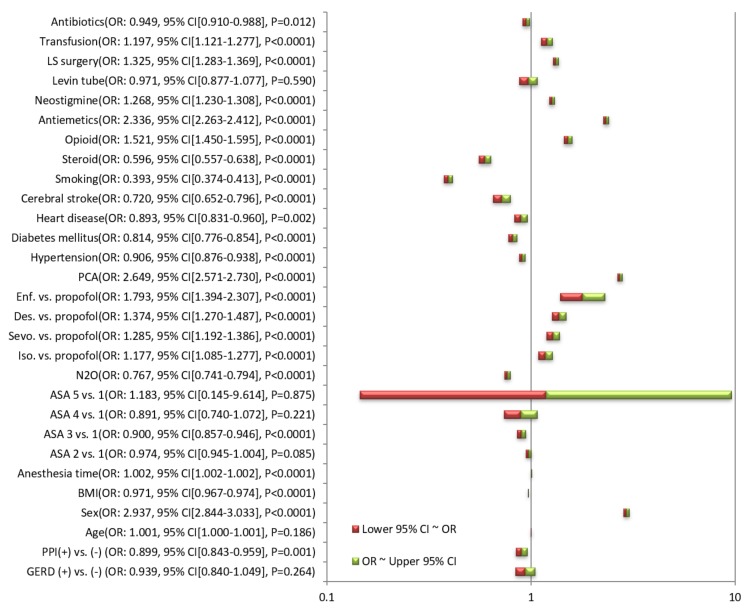
Unadjusted odds ratios for the occurrence of PONV of each variable obtained through logistic regression. GERD, gastroesophageal reflux disease; PPI, proton pump inhibitor; BMI, body mass index; ASA, American Society of Anesthesiologist physical status; Iso., isoflurane; Sevo., sevoflurane; Des., desflurane; Enf., enflurane; PCA, patient-controlled analgesia; LS, laparoscopic; OR, odds ratio; CI, confidence interval; PONV, postoperative nausea and vomiting.

**Figure 2 jcm-09-00825-f002:**
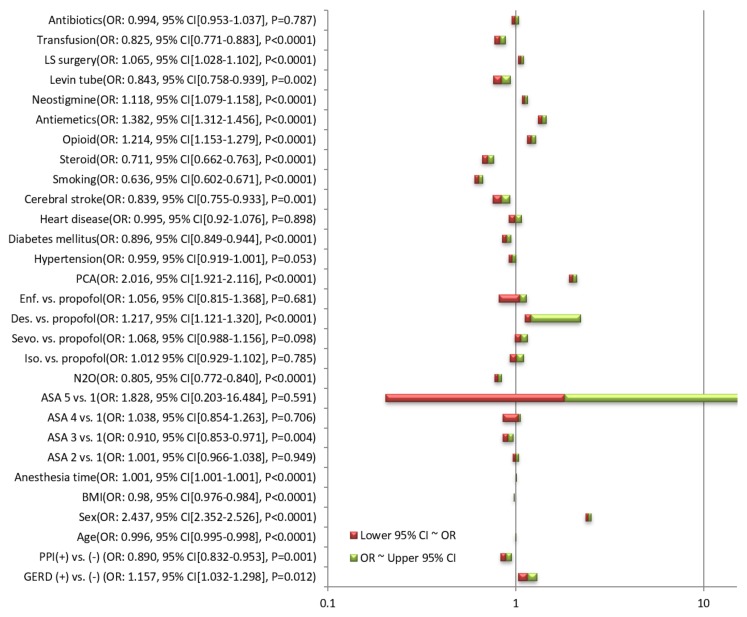
Adjusted odds ratios for the occurrence of PONV of each variable obtained through multivariate logistic regression including all variables. GERD, gastroesophageal reflux disease; PPI, proton pump inhibitor; BMI, body mass index; ASA, American Society of Anesthesiologist Physical Status; Iso., isoflurane; Sevo., sevoflurane; Des., desflurane; Enf., enflurane; PCA, patient-controlled analgesia; LS, laparoscopic; OR, odds ratio; CI, confidence interval; PONV, postoperative nausea and vomiting.

**Table 1 jcm-09-00825-t001:** Demographic characteristics and clinical data of patients undergoing general anesthesia.

		No-PONV (*n* = 181,078)	PONV (*n* = 21,361)	*p*-Value
Mean age (y)		49.2 ± 16.8	49.3 ± 16.1	0.172
Women		90,677 (50.1%)	15,947 (74.7%)	<0.0001
Mean BMI		24.2 ± 3.8	23.8 ± 3.7	<0.0001
Mean anesthesia time (min)		136.7 ± 93.0	154.9 ± 90.9	<0.0001
ASA physical status	1	77,889 (43.0%)	9408 (44.0%)	0.001
	2	82,280 (45.4%)	9680 (45.3%)	
	3	19,731 (10.9%)	2146 (10.0%)	
	4	1171 (0.6%)	126 (0.6%)	
	5	7 (0.0%)	1 (0.0%)	
N2O		46,680 (25.8%)	4494 (21.0%)	<0.0001
Anesthetic maintenance agent	Isoflurane	30,114 (16.6%)	3281 (15.4%)	<0.0001
	Sevoflurane	102,197 (56.4%)	12,156 (56.9%)	
	Desflurane	39,766 (22.0%)	5057 (23.7%)	
	Enflurane	464 (0.3%)	77 (0.4%)	
	Propofol	8,537 (4.7%)	790 (3.7%)	
GERD		3184 (1.8%)	353 (1.7%)	0.264
PPI		10040 (5.5%)	1071 (5.0%)	0.001
PCA		78,560 (43.4%)	14,311(67.0%)	<0.0001
Hypertension		43,951 (24.3%)	4809 (22.5%)	<0.0001
Diabetes mellitus		20,659 (11.4%)	2027 (9.5%)	<0.0001
Heart disease		7983 (4.4%)	845 (4.0%)	0.002
Cerebral stroke		4952 (2.7%)	424 (2.0%)	<0.0001
Smoking		34,950 (19.3%)	1836 (8.6%)	<0.0001
Opioid use during surgery		155,963 (86.1%)	19,316 (90.4%)	<0.0001
Steroid		12,659 (7.0%)	916 (4.3%)	<0.0001
Antiemetics		99,149 (54.8%)	15,780 (73.9%)	<0.0001
Neostigmine use		48,579 (26.8%)	6780 (31.7%)	<0.0001
Levin tube		3638 (2.0%)	417 (2.0%)	0.574
Laparoscopic surgery		38,401 (21.2%)	5616 (26.3%)	<0.0001
Transfusion		7780 (4.3%)	1089 (5.1%)	<0.0001
Antibiotics		157,137 (86.8%)	18,405 (86.2%)	0.012

Postoperative nausea and vomiting; BMI, body mass index; ASA, American Society of Anesthesiologists; GERD, gastroesophageal reflux disease; PPI, proton pump inhibitor; PCA, patient-controlled analgesia. Data are numbers of participants unless otherwise specified.

**Table 2 jcm-09-00825-t002:** Adjusted odds ratios for the occurrence of PONV by GERD and PPI status.

		Odds Ratio (95% CI)	*p*-Value
Adjusted through forward selection of variables	GERD (-)	Reference	
	GERD (+)	1.157 (1.031–1.297)	0.013
	PPI (-)	Reference	
	PPI (+)	0.891 (0.833–0.954)	0.001
Adjusted through backward elimination of variables	GERD (-)	Reference	
	GERD (+)	1.157 (1.032–1.297)	0.013
	PPI (-)	Reference	
	PPI (+)	0.890 (0.832–0.953)	0.001

GERD, gastroesophageal reflux disease; PPI, proton pump inhibitor; CI, confidence interval. Forward selection variables: PPI, GERD, age, sex, BMI, anesthesia time, American Society of Anesthesiologist Physical Status, nitrous oxide, anesthesia maintenance agent, patient-controlled analgesia, diabetes mellitus, cerebral stroke, smoking, steroid, opioid, antiemetics, neostigmine, Levin tube, laparoscopic surgery, and transfusion. Backward elimination variables: PPI, GERD, age, sex, BMI, anesthesia time, American Society of Anesthesiologist Physical Status, nitrous oxide, anesthesia maintenance agent, patient-controlled analgesia, hypertension, diabetes mellitus, cerebral stroke, smoking, steroid, opioid, antiemetics, neostigmine, Levin tube, laparoscopic surgery, transfusion, and antibiotics.

**Table 3 jcm-09-00825-t003:** Types of PPI according to presence and absence of GERD.

	GERD (−) (*n* = 198,902)	GERD (+) (*n* = 3537)
Omeprazole	219 (0.1%)	26 (0.7%)
Pantoprazole	2873 (1.4%)	118 (3.3%)
Lansoprazole	4029 (2.0%)	156 (4.4%)
Rabeprazole	353 (0.2%)	17 (0.5%)
Esomeprazole	3011 (1.5%)	156 (4.4%)
Ilaprazole	113 (0.1%)	4 (0.1%)
Dexlansoprazole	28 (0.0%)	5 (0.1%)

GERD, gastroesophageal reflux disease; PPI, proton pump inhibitor.

**Table 4 jcm-09-00825-t004:** Unadjusted and fully adjusted odds ratios for the occurrence of PONV by PPI type according to presence and absence of GERD.

		GERD (−) (*n* = 198,902)	*p* Value	GERD (+) (*n* = 3537)	*p* Value
Unadjusted	No PPI	Reference		Reference	
OR (95% CI)	Omeprazole	0.664 (0.399–1.104)	0.114	<0.0001	0.998
	Pantoprazole	0.792 (0.694–0.903)	<0.0001	1.137 (0.631–2.048)	0.669
	Lansoprazole	0.884 (0.795–0.984)	0.023	1.350 (0.832–2.192)	0.224
	Rabeprazole	0.986 (0.701–1.386)	0.934	1.224 (0.279–5.380)	0.789
	Esomeprazole	1.033 (0.920–1.159)	0.585	1.198 (0.723–1.986)	0.484
	Ilaprazole	0.390 (0.159–0.955)	0.039	<0.0001	0.999
	Dexlansoprazole	0.312 (0.042–2.295)	0.252	<0.0001	0.999

Fully adjusted	No PPI	Reference		Reference	
OR (95% CI)	Omeprazole	0.819 (0.488–1.373)	0.449	<0.0001	0.998
	Pantoprazole	0.856 (0.748–0.980)	0.025	1.020 (0.540–1.928)	0.951
	Lansoprazole	0.801 (0.718–0.894)	<0.0001	1.019 (0.603–1.722)	0.945
	Rabeprazole	0.890 (0.628–1.262)	0.513	0.643 (0.138–3.001)	0.575
	Esomeprazole	1.085 (0.963–1.223)	0.180	0.982 (0.565–1.706)	0.948
	Ilaprazole	0.391 (0.158–0.966)	0.042	<0.0001	0.999
	Dexlansoprazole	0.355 (0.048–2.645)	0.312	<0.0001	0.999

GERD, gastroesophageal reflux disease; PPI, proton pump inhibitor; OR, odds ratio. When fully adjusted, variables: types of PPI, GERD, age, sex, BMI, anesthesia time, American Society of Anesthesiologist Physical Status, nitrous oxide, anesthesia maintenance agent, patient-controlled analgesia, hypertension, diabetes mellitus, heart disease, cerebral stroke, smoking, steroid, opioid, antiemetics, neostigmine, Levin tube, laparoscopic surgery, transfusion, and antibiotics.

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
