# Peer review of "Effect of a Preoperative Proton Pump Inhibitor and Gastroesophageal Reflux Disease on Postoperative Nausea and Vomiting"

_jcm, 2020, doi:10.3390/jcm9030825_

Round 1

Reviewer 1 Report

Dr. Kwon and colleagues conducted this large retrospective study to assess the effects of PPI and GERD on PONV.

There are some comments with regard to the methodology and the presentation.

1) My major concern is that indications for PPI treatment, drug choice, doses and duration of treatment are not described. This leads to a potential significant heterogeneity of patient characteristics that might influence the results.

2) Even if GERD was assessed using the 10th International Statistical Classification of Diseases and Related Health Problems, details about the diagnosis and treatment are lacking. The reason of this comment is the same as for point 1.

3) Figure legends should explain the colours used.

4) the Discussion section needs extensive rephrasing. In the first paragraph there is a repetition of results. The second paragraph is not clear to follow for the reader: discussion of previous studies should be improved, summarised and clarified. The authors could extend the discussion in order to give an explanation of their findings.

Author Response

Thanks for your thoughtful review of our manuscript. We were deeply impressed by your comments. We take your comments seriously and have addressed them to the best our abilities. As per your suggestions, we have made a revision to our manuscript. We hope you will agree with our answers to your suggested changes. Thank you.

  1. My major concern is that indications for PPI treatment, drug choice, doses and duration of treatment are not described. This leads to a potential significant heterogeneity of patient characteristics that might influence the results.

Answer:  We also agree with your opinion. 

The indications and doses for using PPI in Korea are determined by the Health and Welfare Department. In Korea, it can be used for the treatment of gasrtic ulcer, duodenal ulcer, GERD and Helicobactor pylori. The full dose is used as the therapeutic dose and the half dose is used as the maintenance dose. However, because we investigated data of period during hospitalization before and after surgery, it was not possible for each patient to know exactly how much PPI was used for what purpose. We could not know the duration of taking PPI before admission.  Because PPIs provide protection against the harmful effects of aspiration pneumonia by raising the pH of gastric contents, a PPI is also used as medication before anesthesia. In addition, the dose of PPI required before anesthesia may be full or half dose depending on the drug. Therefore, we did not focus on indications for PPI treatment, drug choice, doses and duration of treatment in this study. We described the indications for taking PPI in method section. Subgroup analysis was conducted according to PPI types. The details that could not be investigated were described in the limitation section.

  1. Even if GERD was assessed using the 10th International Statistical Classification of Diseases and Related Health Problems, details about the diagnosis and treatment are lacking. The reason of this comment is the same as for point 1.

Answer: We also agree with your opinion. In this study, GERD included not only cases of abnormal findings by endoscopy but also those diagnosed with reflux symptoms. In Korea, a full dose can be prescribed only if the endoscopy is done, and only half dose can be prescribed if the diagnosis is based solely on symptoms. Unfortunately, dose may be chosen for institutional reasons. Because it is difficult to know how the patient was diagnosed and the dosage of drugs was determined. Therefore, we only investigated whether PPI was used.

  1. Figure legends should explain the colours used.

Answer: We explained the color used in figure legend.

  1. the Discussion section needs extensive rephrasing. In the first paragraph there is a repetition of results. The second paragraph is not clear to follow for the reader: discussion of previous studies should be improved, summarised and clarified. The authors could extend the discussion in order to give an explanation of their findings.

Answer: We also agree with your opinion. We revised the first and second paragraphs and expanded the discussion with additional content. We have improved the discussion of previous studies, and revised the first and second paragraphs and expanded the discussion with additional content, including new analysis results.

Reviewer 2 Report

The research is interesting, covering a very large number of subjects.

PPIs are inactivated by polymorphic CYPs. PPIs are predominantly metabolized by CYP2C19 and to a lesser extent by CYP3A. Both CYP isoenzymes are highly polymorphic. 

  • Are there any data about the pharmacogenetic status of the patients?
  • Is there any data about the individual dosage of PPIs?
  • How many of the patients were PPI over-, under- or well-treated?

Author Response

Thanks for your thoughtful review of our manuscript. We were deeply impressed by your comments. We take your comments seriously and have addressed them to the best our abilities. As per your suggestions, we have made a revision to our manuscript. We hope you will agree with our answers to your suggested changes. Thank you.

  1. Are there any data about the pharmacogenetic status of the patients?

Answer: Thank you for your good point which we could not think of. Because it does not usually record the pharmacogenetic status for anesthesia patients except in special cases, we could not obtain data on the pharmacogenetic status of patients. However, we also believe that if we get pharmacogenetic data of the patients, this study could yield better results. Instead, we categorized PPIs by type, and analyzed the relationship between the incidence of PONV and each type of PPI according to presence or absence of GERD.

  1. Is there any data about the individual dosage of PPIs?

Answer:  Thank you for your good feedback.

The indications and doses for using PPI in Korea are determined by the Health and Welfare Department. In Korea, it can be used for the treatment of gasrtic ulcer, duodenal ulcer, GERD and Helicobactor pylori. The full dose is used as the therapeutic dose and the half dose is used as the maintenance dose. However, because we investigated data of period during hospitalization before and after surgery, it was not possible for each patient to know exactly how much PPI was used for what purpose. Because PPIs provide protection against the harmful effects of aspiration pneumonia by raising the pH of gastric contents, a PPI is also used as medication before anesthesia. In addition, the dose of PPI required before anesthesia may be full or half dose depending on the drug.

  1. How many of the patients were PPI over-, under- or well-treated?

Thank you for your good question. As mentioned above, we did not know how much PPI was used for, so we could not know how many patients were over-, under- or well- treated.

Round 2

Reviewer 1 Report

The authors have addressed my comments.

I just suggest minor changes to some sentences: e.g. in the discussion

- "Women are well known risk factors for PONV, including our study." --> Female gender is a well known risk factor for PONV, and this is confirmed in our study

  • "GERD patients with taking preoperative PPI" --> GERD patients receiving preoperative PPIs
  • "Ilaprazole showed the lowest probability of occurring PONV" --> was associated with the lowest probability of PONV
  • "Other limitations of our study included a lack of information on dose and duration of PPI" --> also on indications for PPI therapy.

In addition, in the conclusion section, the sentence "Therefore, our findings suggest that a preoperative lansoprazole or pantoprazole or ilaprazole should be taken by patients without GERD undergoing anesthesia and that patients with GERD should consider additional methods to reduce PONV." should be rephrased more cautiously, as RCTs should be expected to give such a conclusion. The authors, in my opinion, could just make a speculation and should call for such studies in order to confirm their opinion.

Author Response

Thanks for your thoughtful review of our manuscript. We were very impressed by your consideration and good advice. We took your comments seriously and revised the manuscript according to your suggestion.

  1. "Women are well known risk factors for PONV, including our study." --> Female gender is a well known risk factor for PONV, and this is confirmed in our study

Answer:  7page 167 line

  • Female gender is a well-known risk factor for PONV, and this is confirmed in our study.

  1. "GERD patients with taking preoperative PPI" --> GERD patients receiving preoperative PPIs

Answer: 7page 180 line

  • GERD patients receiving preoperative PPIs

  1. "Ilaprazole showed the lowest probability of occurring PONV" --> was associated with the lowest probability of PONV

Answer: 7page 186 line

  • Ilaprazole was associated with the lowest probability of PONV

  1. "Other limitations of our study included a lack of information on dose and duration of PPI" --> also on indications for PPI therapy.

Answer: 8page 220-1 line

  • Other limitations of our study also included a lack of on indications for PPI therapy
  1. In addition, in the conclusion section, the sentence "Therefore, our findings suggest that a preoperative lansoprazole or pantoprazole or ilaprazole should be taken by patients without GERD undergoing anesthesia and that patients with GERD should consider additional methods to reduce PONV." should be rephrased more cautiously, as RCTs should be expected to give such a conclusion. The authors, in my opinion, could just make a speculation and should call for such studies in order to confirm their opinion.

Answer: 8page 229 line

  • Therefore, our findings provide additional support that GERD is associated with increasing PONV and preoperative esomeprazole is not effective at reducing PONV. However, future prospective, population-based studies are needed to investigate the association between preoperative use of lansoprazole, pantoprazole and ilaprazole, and its impact on reducing PONV.